# Emulating chemistry-climate dynamics with a linear inverse model

Eric J. Mei<sup>1</sup>, Gregory J. Hakim<sup>1</sup>, Max Taniguchi-King<sup>1,2</sup>, Dominik Stiller<sup>1</sup>, and Alexander J. Turner<sup>1</sup>

Correspondence: Eric J. Mei (emei@uw.edu)

Abstract. Coupled chemistry–climate models (CCMs) are powerful tools for investigating chemical variability in the climate system, but high computational cost limits their use for hypothesis testing and adequately sampling variability on long timescales. Here, we present the first application of a linear inverse model (LIM) to emulate a CCM. A LIM is a low-dimensional empirical model that reproduces the CCM's statistics and dynamics at low computational cost. By linearizing the CCM's dynamics, the LIM captures coherent modes of variability, such as the El Niño Southern Oscillation (ENSO), that describe the coupled evolution of physical and chemical fields. Deterministic seasonal forecasts of the LIM result in skillful predictions of physical and chemical variables at lead times up to a year, outperforming damped persistence models. We show that the LIM's skill in chemical fields depends on its coupled chemistry–climate modes: forecasts without the ENSO dynamical mode show a substantial loss of skill, suggesting the importance of ENSO in driving predictable chemical variability. These results demonstrate that the LIM can efficiently emulate CCM dynamics. It offers a practical tool for testing hypotheses about the drivers of chemistry-climate interactions and may enable efficient chemical data assimilation in the future.

# 1 Introduction

Atmospheric composition is modulated by both external forcing and natural climate variability. Understanding and predicting the underlying drivers of this variability is essential to identify the forced trend in tropospheric oxidants such as the hydroxyl radical (OH; e.g., Prinn et al., 2001; Naik et al., 2013; Chua et al., 2023) and ozone (e.g., Thornhill et al., 2021; Fiore et al., 2022), which determine the lifetimes of trace gases including ozone-depleting substances and methane (e.g., Turner et al., 2019). Ozone itself is also a potent greenhouse gas. Coupled chemistry-climate models (CCMs), general circulation models with interactive tropospheric chemistry, are powerful tools for investigating the mechanisms of chemistry-climate interactions and the magnitude of chemical variability. Long integrations or large ensembles from CCMs enable statistical characterization of the internal variability of the chemical system (e.g., Turner et al., 2018; Fiore et al., 2022; Zhu et al., 2024). However, their computational cost limits applications such as adequately sampling variability or rigorously investigating the dynamics of chemistry-climate interactions. Here, we present a simple empirical model that emulates the statistics and dynamics of CCMs at low computational cost. We use this model to probe the predictability of chemistry-climate variability.

Climate variability drives changes in the chemical system by modulating chemical production, loss, and transport. Multiple modes of variability have been shown to have a significant impact on the variability of the OH and ozone, including the El Niño Southern Oscillation (ENSO; Prinn et al., 2005; Doherty et al., 2006; Turner et al., 2018; Rowlinson et al., 2019; Anderson

<sup>&</sup>lt;sup>1</sup>Department of Atmospheric and Climate Science, University of Washington, Seattle, WA 98195, USA

<sup>&</sup>lt;sup>2</sup>Department of Earth and Planetary Sciences, University of California, Berkeley, CA 94720, USA

et al., 2021; Zhu et al., 2024) and the Indian Ocean Dipole (Anderson et al., 2021, 2024). El Niño produces a positive-negative ozone dipole in the west and east Pacific, respectively, mostly due to changes in the Walker circulation (Doherty et al., 2006). El Niño produces a negative global OH anomaly due to reduced lightning  $NO_x$  emissions, which decrease  $HO_x$  recycling (Murray et al., 2013; Turner et al., 2018), and increased CO biomass burning emissions, which increase OH loss (Prinn et al., 2005; Rowlinson et al., 2019; Zhao et al., 2020). While global OH concentrations decrease during El Niño, Anderson et al. (2021) note an increase in OH in the planetary boundary layer due to increases in water vapor and the primary production of OH.

CCMs can simulate the response of chemical species to climate modes of variability, but their computational expense limits their use in investigating chemistry-climate dynamics or in directly reconstructing historical chemical variability with data assimilation. For these applications, past work has often relied on non-dynamical diagnostic or statistical methods that infer relationships from coincident correlations between variables or their spatial patterns. Empirical orthogonal function analysis on CCM simulations has been used to estimate the magnitude of OH variability caused by ENSO and to hypothesize its underlying drivers (Turner et al., 2018; Anderson et al., 2021). Machine learning models trained on CCM outputs have been used to diagnose differences in OH variability across models (Nicely et al., 2020) and to infer historical OH variability from satellite observations of related chemical species (Anderson et al., 2022, 2023, 2024; Zhu et al., 2024). Non-dynamical techniques remove the system's feedbacks and memory, which are essential for capturing lagged or emergent responses to perturbations in the chemical system (e.g., Prather, 1994; Zhao et al., 2025). As a result, it is difficult to determine the causal impact of climate variability on chemical species without a model that includes time-dependent behavior.

45

We present the linear inverse model (LIM) as a computationally efficient, dynamical emulator of CCMs. The LIM is a low-dimensional empirical model calibrated on observational or model simulation data (Penland and Sardeshmukh, 1995). Prior applications have focused on physical climate dynamics and provide a framework for using the LIM to investigate chemistry-climate variability. The LIM has been used to understand the structure, evolution, and dynamical influence of climate modes such as ENSO (Penland and Sardeshmukh, 1995; Vimont et al., 2014, 2022) and the Pacific Decadal Oscillation (Alexander et al., 2008), where slow linear dynamics dominate. The simple, linear formulation of the LIM allows easy modification of its components to probe dynamical relationships between variables in initial value problems or free-running simulations (Vimont et al., 2014) or use of linear systems analysis to investigate the growth structure of dynamical modes (Penland and Sardeshmukh, 1995; Alexander et al., 2008; Vimont et al., 2022). As forecast models, the LIM performs comparably to general circulation models from the CMIP5 generation at predicting observed ocean and atmospheric temperatures for lead times of up to 9 years (Newman, 2013), and thus have been used to assess predictability of coupled atmosphere-ocean states (Alexander et al., 2008; Huddart et al., 2017; Perkins and Hakim, 2020) and sea ice (Brennan et al., 2023). The LIM's computational efficiency enables its use as a forward model in coupled, online reconstructions of unobserved past climate states (e.g., ocean and atmosphere) over the last millennium (Perkins and Hakim, 2021).

The remainder of the paper is organized as follows. In Section 2, we present LIM theory and practical application to existing CCM simulation data. Section 3 validates the statistics of an ensemble of long LIM simulations and investigates the coupled

chemistry–climate dynamical modes of the LIM. Section 4 assesses the predictability of chemistry-climate variability, and conclusions are provided in Section 5.

## 2 Calibrating a linear inverse model (LIM) on the chemistry-climate system

## 2.1 LIM theory

The linear inverse model (LIM; Penland and Sardeshmukh, 1995) is a low-dimensional dynamical representation of a statistically stationary dynamical system:

$$\frac{dx}{dt} = \mathbf{L}x + \xi. \tag{1}$$

The LIM decomposes the tendency of anomalies, dx/dt, about a time-independent mean state into a linear, deterministic operator L and a stochastic process  $\xi$  that is uncorrelated in time but may be correlated in the state. The anomalies of any variable from the original dynamical system may comprise the state vector x with the assumption that the variable's dynamics can be represented by slow-varying, linear dynamics and fast-timescale Gaussian noise representing nonlinearity and unresolved processes.

To produce a forecast from state x(t) over any time increment  $\tau$ , Eq. 1 is integrated and represented in discrete time:

$$x(t+\tau) = \mathbf{G}(\tau)x(t) + \zeta(t,\tau). \tag{2}$$

The deterministic evolution of the state x(t) from time t to time  $t + \tau$  is represented by the propagation matrix  $\mathbf{G}(\tau)$ , which is the integral of  $\mathbf{L}$  over  $\tau$ :  $\mathbf{G}(\tau) = \exp(\mathbf{L}\tau)$ . The integral of the stochastic process over  $\tau$  is represented by  $\zeta(t,\tau)$ .

To calibrate the deterministic propagation matrix, we define the time-lag ( $\tau$ -lag) covariance of a system as  $\mathbf{C}(\tau) = \langle \boldsymbol{x}(t+\tau)\boldsymbol{x}(t)^{\mathsf{T}}\rangle$ , in which  $\langle \cdot \rangle$  denotes an expectation (taken as a time-average here). Given a trajectory of the original dynamical system sampled at time increments of  $\tau_0$ , we calibrate the  $\tau_0$ -lag propagation matrix  $\mathbf{G}(\tau_0)$  by dividing the  $\tau_0$ -lag covariance of the system with the 0-lag covariance:

$$\mathbf{G}(\tau_0) = \mathbf{C}(\tau_0)\mathbf{C}(0)^{-1}.$$

The calibration of the propagation matrix is akin to a multivariate least-squares linear regression between two time steps. In essence,  $\mathbf{G}(\tau_0)$  provides the most likely linear evolution of the state over the time increment  $\tau_0$  in a least-squares sense.  $\mathbf{L}$  can be recovered from  $\mathbf{G}(\tau_0)$  by  $\mathbf{L} = \tau_0^{-1} \ln \mathbf{G}(\tau_0)$ , which gives a continuous representation of the linear tendency derived from the discrete data (Eq. 1). In theory,  $\mathbf{L}$  should be insensitive to the time resolution  $\tau_0$  of the training data (Penland and Sardeshmukh, 1995); this is known as the "tau test." In practice, LIMs calibrated on the same data at different temporal resolutions may fail the tau test due to sampling error, a violation of the LIM's assumptions, or insufficient temporal resolution of important dynamical modes (Penland, 2019).

The LIM is statistically stationary, meaning the statistics of the system averaged over a large sample trajectory do not change over time. For the state to remain bounded in time, L must be damped such that the real parts of its eigenvalues are

all negative. The noise forcing  $\boldsymbol{\xi}$  is required to maintain the covariance statistics for long integrations. We model this forcing as a Gaussian temporally-white process with intensity  $\mathbf{Q}$  such that the covariance of this forcing over a small time interval dt is  $\mathbf{Q}dt = \langle (\boldsymbol{\xi}dt)(\boldsymbol{\xi}dt)^{\mathsf{T}} \rangle$ .  $\mathbf{Q}$  is derived from the fluctuation-dissipation relationship (Penland and Matrosova, 1994), which relates the forcing intensity to  $\mathbf{C}(0)$  and  $\mathbf{L}$ :

$$\frac{d\mathbf{C}(0)}{dt} = \mathbf{L}\mathbf{C}(0) + \mathbf{C}(0)\mathbf{L}^{\mathsf{T}} + \mathbf{Q} = 0.$$
(4)

The covariance of this noise represents fast-timescale processes that the linear dynamics cannot resolve (e.g., nonlinearities, unrepresented processes, etc.).

Practically, LIM integration must be performed carefully because of the LIM's temporally uncorrelated stochastic component. We integrate the LIM with the two-step integration scheme posed by Penland and Matrosova (1994), which generates temporally-consistent states that preserve the spatial and spectral statistics of the system over long integrations. This scheme has been used extensively to integrate LIMs over millennial timescales in previous applications (e.g., Perkins and Hakim, 2021):

$$y(t + \delta t) = (\mathbf{I} + \mathbf{L}\,\delta t)x(t) + \hat{\mathbf{Q}}\sqrt{\mathbf{\Lambda}_Q\,\delta t}\,\boldsymbol{\alpha}$$
(5)

$$\boldsymbol{x}(t+\delta t/2) = [\boldsymbol{x}(t) + \boldsymbol{y}(t+\delta t)]/2. \tag{6}$$

Here,  $\boldsymbol{y}$  is an intermediate variable used to update  $\boldsymbol{x}$ . For a small time step  $\delta t$  (e.g., 4 hours in this work), the deterministic component of  $\boldsymbol{y}$ ,  $(\mathbf{I} + \mathbf{L} \delta t) \boldsymbol{x}(t)$ , arises from a forward-Euler approximation of the linear tendency. The noise forcing  $\hat{\mathbf{Q}} \sqrt{\Lambda_Q \delta t} \alpha$  comes from an eigen-decomposition of the noise forcing intensity  $\mathbf{Q} = \hat{\mathbf{Q}} \Lambda_Q \hat{\mathbf{Q}}^{-1}$ , in which the eigenvectors are the columns of  $\hat{\mathbf{Q}}$  and eigenvalues are the diagonal elements of  $\Lambda_Q$ .  $\alpha$  is a vector of elements sampled independently from a standard normal distribution. One issue with constructing the noise forcing is that the noise forcing covariance matrix may not be positive definite. If the negative eigenvalues comprise a small fraction of the total stochastic variance (e.g., the negative  $\mathbf{Q}$  eigenvalues in our monthly LIM comprise less than 5% of the total  $\mathbf{Q}$  variance), these offending eigenvalues and their associated eigenvectors may be removed, and the remaining positive eigenvalues proportionally scaled such that their sum matches the trace of  $\mathbf{Q}$  (Perkins and Hakim, 2021). Similar to prior studies (e.g., Perkins and Hakim, 2021), we find that the proportion of negative eigenvalues to the trace of  $\mathbf{Q}$  is sensitive to our data pre-processing, though no coherent relationships were found (Figure S1 in the Supplemental).

#### 2.2 LIM calibration on chemistry-climate model data

100

To calibrate and test our LIM, we use a 6,000-year pre-industrial control simulation from the Geophysical Fluid Dynamics Laboratory Climate Model version 3 (GFDL-CM3; Donner et al., 2011) that was previously used by Turner et al. (2018). Briefly, GFDL-CM3 is a general circulation model with interactive atmosphere, ocean, land, and sea ice components (Griffies et al., 2011; Naik et al., 2013). The atmospheric model uses a cubed sphere configuration with  $48 \times 48$  cells per face and 48 vertical pressure levels (1 Pa at model top) for the dynamical core (Donner et al., 2011), resulting in a  $2^{\circ} \times 2.5^{\circ}$  horizontal resolution. Atmospheric composition is interactive, in which the tropospheric chemistry scheme is based on MOZART-2 (Model

for OZone and Related chemical Tracers; Horowitz et al., 2003), and the stratospheric chemistry scheme is based on AM-TRAC (Atmospheric Model with Transport and Chemistry; Austin and Wilson, 2006). Land cover, climate forcings, greenhouse gas surface concentrations, and surface emissions of carbon monoxide (CO), nitric oxide (NO), and organic compounds are prescribed in annually-repeating seasonal cycles at 1860 conditions. Lightning emissions of nitrogen oxides (NO $_x$ ) are calculated interactively with a convective cloud-top height scheme. Outputs from the simulation are saved in monthly-averaged increments. We use the stable last 3,000 years of the simulation, from which we use 2,000 years to calibrate our LIM and withhold 1,000 years to test our LIM.

150

155

We build the LIM state vector with variables that (1) we want to predict or provide predictability, and/or (2) may explain the physical or chemical processes of the phenomena being investigated. From the GFDL-CM3 simulation, we calibrate our LIM on a subset of variables that form a minimal representation of the coupled chemistry-climate system: we use sea surface temperatures (SST) to represent physical climate dynamics, and we use atmospheric concentrations of OH, CO,  $NO_x$ , and ozone ( $O_3$ ) for a simple representation of chemical dynamics that modulate global concentrations of OH (e.g., Murray et al., 2013, 2021). Our conclusions are not sensitive to the subset of chemical variables used. While we train and test the LIM on all aforementioned variables, we show results for SST, OH, and ozone in the main text (see Supplemental Section 2 for the remaining variables). We choose to highlight OH and ozone to demonstrate LIM performance for both short- and longer-lived chemical species, respectively.

Following Turner et al. (2018), mass-weighted averages over the free tropospheric column (850 to 300 hPa) are computed for each chemical species in each horizontal grid cell. Conclusions are largely insensitive to the choice of free tropospheric or entire tropospheric column averages (not shown). Data are de-trended, and the seasonal cycle for each variable at each grid cell is removed by subtracting the time-averaged value for each month, which transforms the data into anomalies about a mean seasonal cycle. However, while the seasonal cycle *mean* has been removed, many variables retain a seasonal cycle in their covariance, particularly in higher latitudes where there are large seasonal changes to insolation. Large changes in the amount of sunlight affect the dynamics of many chemical species, which results in considerable differences in the magnitude of anomalies between seasons. For example, OH production by ozone photolysis is negligible in the winter for high latitudes, which results in near-zero anomalies. This seasonal cycle in variance is potentially problematic for our methodology because we calibrate one LIM to emulate dynamics across all seasons. Here we choose to mask out anomalies of all chemical states in polar latitudes (>60°), however future work could calibrate one LIM for each month or season (e.g., Shin et al., 2021). OH requires a more stringent mask because its production is strongly dependent on sunlight in the midlatitudes. We mask out all data in the extratropics for OH (>30°). Despite this large spatial mask, only  $\sim$ 20% or less of global area-weighted variance in the chemical fields is removed by this masking. This can be seen in Figure 1, which shows the grid cell variance for ozone, OH, and SSTs. For SST, changes to sea ice extent causes large interannual-to-decadal variability in the poles. As such, we again mask out polar regions ( $>60^{\circ}$ ) for SST, which removes relatively more variance (57%) compared to the chemical data.

In order to effectively use limited training data, a LIM typically employs a low-dimensional representation of the system's state prior to calibration. We compress the state using empirical orthogonal functions (EOFs) of each masked variable. EOFs provide an orthogonal basis of a variable in which the leading EOFs optimally explain the variance in the data. To account for

**Figure 1.** Variance remaining after latitude masking and EOF truncation. (a) Cumulative percentage of global field variance retained as a function of the number of EOFs in the LIM state vector. Dashed lines indicate percentage of global field variance remaining after high latitudes are masked out. Percentage of global field variance in each grid cell for (b) ozone, (c) OH, and (d) SST. Dashed lines indicate latitude cut-off for the extratropical masks.

differing areas of grid cells in the model data, we weight the data by latitude ( $\sqrt{\cos(\text{lat})}$ ). Defining the weighted data matrix for a single variable to be  $\mathbf{Z} \in \mathbb{R}^{m \times n}$ , where m is the number of grid points (gridded anomalies stacked in a vector) and n is the number of time samples, we compute the variable's EOFs  $\Psi$  using a singular value decomposition:  $\mathbf{Z} = \Psi \Sigma \Phi^{\mathsf{T}}$ . Then, we retain the leading p EOFs (here, p = 40) and truncate the remainder:  $\tilde{\Psi} \in \mathbb{R}^{m \times p}$ . The leading 40 EOFs explain 54% to 64% of the global field variance of the chemical data (Figure 1). Retaining more EOFs per variable does not notably change our results. We project the weighted data onto the retained EOFs by  $\tilde{\mathbf{Z}} = \tilde{\Psi}^{\mathsf{T}} \mathbf{Z}$ ;  $\tilde{\mathbf{Z}} \in \mathbb{R}^{p \times n}$ . Finally, we standardize the projected data by the total retained field variance  $\sigma^2$  to ensure that variables have comparable variance when calibrating the LIM (Perkins and Hakim, 2020). We perform this process for each variable and concatenate the resulting matrices to form the final data matrix for the LIM (Eq. 7). Columns of  $\mathbf{X}$  are LIM states  $\boldsymbol{x}$  at equally-spaced time samples, which is used for training using Eq. 3:

$$\mathbf{X} = \begin{bmatrix} \mathbf{\tilde{Z}}_{\text{SST}}^{\mathsf{T}}, \ \mathbf{\tilde{Z}}_{\text{OH}}^{\mathsf{T}}, \ \mathbf{\tilde{Z}}_{\text{O_3}}^{\mathsf{T}}, \ \mathbf{\tilde{Z}}_{\text{O_3}}^{\mathsf{T}}, \ \mathbf{\tilde{Z}}_{\text{CO}}^{\mathsf{T}}, \ \mathbf{\tilde{Z}}_{\text{NO}_x}^{\mathsf{T}} \end{bmatrix}^{\mathsf{T}} \in \mathbb{R}^{5p \times n}$$

$$(7)$$

## 3 LIM emulation of CCM statistics and dynamics

200

A well-calibrated LIM reproduces the spatial and temporal statistics of the original dynamical system. We calibrate a LIM on 2,000 years of monthly data from the GFDL-CM3 simulation. We use the remaining 1,000 years of data to validate the 170 statistics of our calibrated LIM. To ensure a one-to-one comparison, we de-trend and de-seasonalize the test data before projecting it onto the same EOF basis used to prepare the LIM training data. We simulate a 300-member ensemble of freerunning 1,000-year integrations of the coupled chemistry-climate system using our LIM. Given the chaotic nature of the climate system, simulations from the LIM and GFDL-CM3 will have different trajectories but should be statistically indistinguishable. 175 Timeseries of tropical OH anomalies from the test data and one member of the LIM ensemble are shown in Figure 2a. One hundred years of simulation with the LIM takes 16.6 s  $\pm$  0.017 s (mean  $\pm$  standard deviation over 10 ensemble members) when run on a single 2.9 GHz core of an Intel Xeon Gold 6226R processor. From Figure 2b, we can see that these timeseries are statistically indistinguishable, producing similar ranges and distributions of tropical OH anomalies over time. To compare the spectral characteristics of the simulations, we compare the power spectral density of tropical OH anomalies of each 1,000-180 year simulation (Figure 2c) using the multitaper method (Prieto et al., 2009). The large 95% confidence interval of the power spectral density of the LIM simulations captures the broad features of the GFDL-CM3 spectrum. The GFDL-CM3 spectrum is noisy given its single realization, especially at high frequencies, but the power spectral density of individual LIM ensemble members show similar noisy characteristics (Figure S24). The LIM may have too little power at the lowest frequencies, but the low frequency portion of the GFDL-CM3 spectrum has few degrees of freedom and is poorly resolved given the length of the sample. Finally, we compare the spatial patterns of standard deviations of OH anomalies across time from one member of the LIM ensemble (Figure 2e) to those from the test data (Figure 2d). The LIM's spatial statistics are comparable to those from GFDL-CM3 (domain-average error of 3.2% for the standard deviation of OH), in which regions of high variance and low variance match between the simulations from the GFDL-CM3 model and the LIM.

To investigate if the LIM captures the dynamics of GFDL-CM3, we decompose the deterministic dynamics of the LIM 190 (L) through eigenanalysis (Eq. 8) into a linear basis of empirical normal modes (ENMs; the column vectors v that comprise  $\mathbf{V}$ ):

$$\mathbf{L} = \mathbf{V} \mathbf{\Lambda} \mathbf{V}^{-1}. \tag{8}$$

ENMs are dynamical modes that oscillate and decay at characteristic timescales. These timescales are recovered from the eigenvalues associated with each ENM (diagonal elements of  $\Lambda$ , which take the form  $\lambda = a + ib$ ), in which the real part of the eigenvalue describes the damping time or "e-folding" time of the ENM,  $\tau = -a^{-1}$ , while the imaginary part of the eigenvalue describes the period of oscillation of the ENM,  $T = 2\pi b^{-1}$ . ENMs must all decay (a 

**Figure 2.** Comparison of tropical OH statistics of 1,000-year free-running integrations of GFDL-CM3 and the LIM. (a) Timeseries of OH averaged over the tropics (between 30° N and 30° S) from GFDL-CM3 (blue) and one LIM simulation (orange). (b) Distribution of tropical OH anomalies over time. (c) Power spectral density of tropical OH anomalies from GFDL-CM3 (blue) and a 300-member ensemble of simulations from the LIM (orange). Solid orange line is the LIM ensemble median, and the shading represents the LIM 95% confidence interval. Standard deviation of OH anomalies over time for the (d) GFDL-CM3 simulation and (e) one simulation from the LIM.

One such oscillating ENM that prior work has identified to be important for physical climate dynamics is an ENM that resembles ENSO (e.g., Penland and Sardeshmukh, 1995; Vimont et al., 2014, 2022). Given that ENSO causes significant OH variability in GFDL-CM3 (Turner et al., 2018), we search for an "ENSO mode" in the ENMs of our LIM. The spatial evolution of an ENM over time can be visualized by rotating its vector in the complex plane, from which only the real component of the ENM appears in the state space. A single state element  $v_j$  of ENM v takes the form  $v_j = c_j + id_j = |v_j| \exp(i\theta_j)$ . The evolution of the real component of the state element is  $\Re\{v_j(t)\} = |v_j| \exp(at) \cos(bt + \theta_j)$ . We identify an ENM with a damping time of 1.3 yr and an oscillation period of 3 yr that strongly resembles ENSO (Figure 3). Spatial patterns of SST anomalies in

the high-variance phases of this ENSO mode are highly correlated with the SST pattern obtained by regressing the Niño 3.4 index onto SST (r > 0.95). To highlight the oscillatory component of the ENSO mode, we show its spatial evolution without damping, and we linearly scale the mode's anomalies such that the variance in the Niño 3.4 index of the training data and the ENSO mode are identical. The ENSO mode captures the evolution of SST through El Niño and La Niña events. During the El Niño stage of the mode, weak positive SST anomalies in the Central Pacific strengthen and propagate eastward, which culminate in a mature El Niño event. The mature El Niño eventually decays and leads to the initial evolution of the La Niña event. The dynamics of La Niña are symmetric to those of El Niño in the ENSO mode.

Because our LIM is calibrated on coupled chemistry-climate dynamics, the ENSO mode captures the co-evolution of the chemical system with physical climate variables. In particular, the ENSO mode captures the relationship between ENSO and OH as noted by Turner et al. (2018), in which the dominant spatial pattern of variability in tropical OH anomalies co-varies with ENSO strength: the principal component of the first OH EOF is almost perfectly out of phase with the Niño 3.4 index (r = -0.98). Consistent with prior work (e.g., Doherty et al., 2006), the ozone anomalies during a mature El Niño exhibit a dipole pattern in the tropical Pacific, in which negative anomalies exist over the central and east Pacific and positive anomalies form a horseshoe shape centered over the Maritime Continent and Indian Ocean. In addition to these expected 0-lag statistics, the LIM's ENSO mode captures the *dynamics* of the chemical system during ENSO. That is, anomalies of chemical variables grow and decay through the phases of ENSO in spatial patterns distinct from their patterns during mature El Niño or La Niña events (i.e., compare the left and right columns of Figure 3 to the center column). Coupled dynamics in the ENSO mode are supported by lead-lag analysis of the independent GFDL-CM3 test data (Figure S25), in which the Niño 3.4 index is shifted in time and linearly regressed with each variable. The patterns of El Niño growth, peak, and decay in the ENSO mode and the lead-lag analysis match for all variables. This is expected, since the deterministic dynamics of the LIM are calibrated with a linear regression of the time-lagged data. The existence of the ENSO mode in the LIM suggests that linear dynamics can capture the dominant drivers of coupled climate-chemical variability. These dynamics are captured for all chemical species regardless of their lifetime in the atmosphere, as both OH (lifetime  $\approx 1$  s) and ozone (lifetime  $\approx 3$  wk) show coherent dynamics with the evolution of SST. Future CCM experiments could be performed to validate the ENSO chemistry-climate dynamics captured by the LIM.

While the ENSO mode shows dynamics consistent with expectations, the dynamics of each individual ENM of the LIM are not guaranteed to be interpretable. Because the eigenanalysis yields an ENM for each element in the state vector, a single dynamical mode could be split among different ENMs depending on how the state vector is structured (e.g., choices in latitude/longitude masking or dimensional reduction). For this reason, it can be difficult to isolate the dynamics of other modes of internal variability in individual ENMs. The ENSO mode we show is particularly robust: the oscillation period and SST dynamics of this mode are insensitive to choices in the LIM calibration such as the pressure bounds of the tropospheric column averaging, the variables chosen for the state vector, the number of EOFs retained, or the time resolution of the training data (monthly or seasonal; not shown).

**Figure 3.** Evolution of the oscillatory component of the ENSO mode without damping. (*top*) Timeseries of the Niño 3.4 index (blue) and the first principal component of OH (orange) as a function of phase. Three vertical black lines indicate three phases during the El Niño stage of the ENSO mode that are visualized in contour plots (*bottom*). From left to right, the columns of contour plots show snapshots of the growth, peak, and decay of anomalies in the phases of the ENSO mode. From top to bottom, the rows of contour plots show SST, OH, and ozone anomalies.

# 4 Predictability of chemistry-climate dynamics with the LIM

To evaluate the performance of the LIM as a forecast model, we use a 300-year independent test dataset from the GFDL-CM3 simulation to perform and validate LIM forecasts. We perform 1-year deterministic forecasts initialized at each state in the

300-year dataset. A deterministic LIM forecast at lead time  $\tau$  that was initialized with initial condition x(0) evolves with the trajectory  $\boldsymbol{x}(\tau) = \mathbf{G}(\tau_0)^{\tau/\tau_0} \boldsymbol{x}(0)$ . Recall that  $\mathbf{G}(\tau_0)$  is the propagation matrix calibrated on data sampled in time increments 245 of  $\tau_0$ . Here, we calibrate our LIM at a seasonal resolution to prolong its forecast skill at long time horizons. We average the original monthly GFDL-CM3 data by season (December-February, March-May, June-August, and September-November). Notably, while this formulation of the LIM produces a fully-damped L operator, variance in its free-running simulations are biased low around 30% compared to the test data due to large negative Q eigenvalues (>10% eigenvalue rescaling necessary). While removing negative eigenvalues and inflating positive eigenvalues preserves the trace of Q, the scaled noise forcing on the remaining Q modes (eigenvectors) does not preserve the equilibrium variance. Predictability in the LIM depends only on the deterministic L operator, but inaccurate stochastic forcing could compromise applications that rely on uncertainty quantification (e.g., data assimilation).

To provide a comparison forecast model to the LIM, we calibrate a damped persistence model  $x(\tau) = \mathbf{A}(\tau_0)^{\tau/\tau_0} x(0)$ , in which  $A(\tau_0)$  is a diagonal matrix comprised of the  $\tau_0$ -lag autocorrelations of the training data in EOF-space used to calibrate 255 the LIM (in Eq. 7). In contrast to the LIM, a damped persistence model has no interaction between state elements: forecasts made with damped persistence decay towards zero anomalies (climatology) at characteristic timescales.

We use the coefficient of efficiency (CE; Nash and Sutcliffe, 1970) as a metric to evaluate the forecast skill of the LIM and the damped persistence model. Given an ensemble of forecasts  $f_i$  and validation data  $v_i$  with N ensemble members, CE is a skill metric that normalizes the mean square error of the ensemble of forecasts relative to the validation data with the mean square error of the climatological forecast  $\bar{v}$  relative to the validation data (Eq. 9). Therefore it penalizes an ensemble of forecasts for (1) bias in the mean of the forecasts, (2) bias in the variance of the forecasts, and (3) lack of correlation between the forecasts and validation:

$$CE = 1 - \frac{\sum_{i=1}^{N} (v_i - f_i)^2}{\sum_{i=1}^{N} (v_i - \bar{v})^2}.$$
(9)

265 A CE of one indicates perfect forecast skill, a CE of zero indicates forecast skill equivalent to climatology, and a negative CE indicates forecast skill worse than climatology (bias in mean or variance). Because CE places forecast error in the context of climatological error variance, we show results in CE rather than in correlation (r) or root mean square error (RSME) in the main text (see Supplemental Section 3 for these other metrics). Conclusions are invariant to the choice of metric used. We mask the high latitude test data for validation but do not project the test data onto the truncated EOF basis used by the LIM.

The LIM outperforms damped persistence for all variables at every forecast lead time (Figure 4). At initialization, the domain-averaged CE for both the LIM and damped persistence models are less than one because the EOF-truncated states have less variance than the untruncated validation data (CE < 0.8 at initialization). Damped persistence has low skill for SST in the first six months of the forecast and is indistinguishable from climatology by nine months. Its forecast skill for the chemical system is even lower, with minimal skill by three months for OH (CE=0.02) and six months for ozone (CE=0.01). In contrast, the LIM exhibits high forecast skill for SST for long (>1 y) lead times (CE decreases from 0.37 to 0.10 from 3 months to 1 year). Forecast skill for chemical variables is lower than for SST but reasonable for prediction of atmospheric variables (e.g., comparable to 2m air temperature reconstruction skill in Perkins and Hakim, 2017): some chemical forecast skill remains for

**Figure 4.** Domain-averaged forecast skill (CE) over lead time from the LIM (red), the LIM with no ENSO mode (purple), and damped persistence (cyan) for (a) SST, (b) OH, and (c) ozone. Dotted horizontal line indicates CE of climatology (CE=0).

forecasts one year after initialization (OH CE decreases from 0.17 to 0.05 from 3 months to 1 year). Skill for seasonal average anomalies of chemical variables is not related to the mean lifetime of the species. For example, forecast skill for OH and CO are comparable (CE=0.17 and 0.15 at 3 months, respectively) despite the much longer lifetime of CO (~12 wk) compared to OH (~1 s). Similar forecast skill for the two species probably occurs because surface emissions of CO are fixed to a seasonal cycle in the GFDL-CM3 simulation, so CO anomalies are driven by changes in transport, chemical production, or loss with respect to OH. Therefore, predictability in the chemical sources and sinks constrained by chemistry-climate interactions is likely more important than atmospheric lifetime in determining forecast skill.

To investigate the difference in skill between the LIM and damped persistence forecasts, we analyze spatial patterns of CE for OH forecasts (Figure 5). Damped persistence loses nearly all of its skill in the first three months of the forecast, where there is only weak skill in regions with high variance in the first EOF of OH. At three months, LIM forecasts have much higher skill than damped persistence, where forecasts exhibit the highest skill over the tropical Pacific and Maritime Continent. The LIM displays moderate skill over the rest of the domain except for over subtropical Africa. As lead time increases, LIM forecast skill decreases but maintains a similar spatial pattern to the three-month forecast. Strong performance of the LIM compared to damped persistence highlights the importance of ENM dynamics in predicting the state of the chemical system. Unlike the LIM, the damped persistence propagation matrix  $\mathbf{A}(\tau_0)$  is fully diagonal, so its ENMs are orthogonal and have no oscillatory component. Forecasts made with damped persistence simply decay with the same spatial pattern as the initial condition. The LIM's non-orthogonal, oscillating ENMs allow an initial condition to grow and evolve its spatial anomalies, following a trajectory closer to the true dynamics of the system. For example, an initial condition in the early stages of El Niño could grow into a mature El Niño if forecast by the LIM but would erroneously decay if forecast by damped persistence.

Regions of high skill in the LIM's OH forecast are coincident with regions of large OH anomalies during the mature phases of the ENSO mode. To investigate the contribution of the ENSO mode towards forecast skill, we project LIM forecasts onto the ENMs, remove the projection onto the ENSO mode, and project the altered forecasts back into the state space for validation (middle column of Figure 5). Removal of the ENSO mode results in lower, sometimes negative, CE at initialization in the highest-variance regions of the ENSO mode. Low skill in these regions are the result of low bias in variance of forecasts with no ENSO mode. At three months, LIM forecasts without the ENSO mode show regions of moderate skill throughout the domain with regions of negative skill over parts of the Maritime Continent, over the tropical east Pacific, and Africa. Skill in all regions, both positive and negative, decay to zero as lead time increases. For all variables, LIM forecasts without the ENSO mode have less than half the domain-averaged skill of the full LIM for lead times out to one year (Figure 4). ENSO dynamics contribute to a large portion of the predictability in the pre-industrial chemical system, which suggests that the chemistry-climate dynamics of ENSO could provide a strong constraint on a reconstruction or prediction of chemical states. Notably, the LIM forecast without the ENSO mode retains regions of positive skill that are absent in the damped persistence forecast, suggesting other sources of dynamic predictability beyond ENSO on sub-annual timescales.

Given the importance of physical climate dynamics for chemical predictability, we repeat our forecast tests with a LIM calibrated on only chemical variables (OH, ozone,  $NO_x$ , and CO) to test the sensitivity of chemical forecast skill to the use of SST during LIM calibration (see Supplemental Section 4). The chemical-only LIM produces OH forecasts that have slightly less skill than the original LIM (OH CE decreases from 0.16 to 0.06 over a year). This result suggests either that the chemical system has its own internal dynamics that are predictable or that the chemical data from the simulation implicitly retains information about physical climate in the absence of any physical climate variables. The latter explanation is more likely, as the chemical-only LIM retains an ENSO mode with similar oscillation period and chemical dynamics as the original LIM (albeit, with a shorter damping time of 0.7 yr compared to 1.3 yr for the original LIM). The drivers of chemical variability are entirely tied to physical climate variability in the pre-industrial control simulation, so the statistics of the chemical data are dominated by physical climate. Lack of sensitivity of the LIM to the physical variable used suggests that any physical

**Figure 5.** OH forecast skill (CE) over lead time from the LIM (*left column*), the LIM with no ENSO mode (*middle column*), and damped persistence (*right column*). Lead time increases from 0 months (*top row*) to 9 months (*bottom row*).

variable that has information about internal variability could be added to the state vector for the chemistry-climate system. This flexibility could be useful for applications of the LIM that require covariance statistics between specific climate fields and the chemical system.

Our predictability experiments assume a system with only slow-varying, natural processes (i.e., internal modes of variability), which we have shown can be captured by the LIM. For the chemical system, external forcings could be substantial (e.g., anthropogenic emissions during industrialization) and short-lived given the short atmospheric lifetime of many chemical species. A fundamental assumption of the LIM is a timescale separation between the linear, resolved dynamics and the fast-timescale, unresolved dynamics (Penland and Sardeshmukh, 1995). External forcings that have similar timescales or dynamics to the resolved dynamics could challenge LIM calibration, and external forcings with unexpected dynamics to the LIM's resolved dynamics could challenge the LIM's predictive skill. Future work should assess the performance of the chemistry-

climate LIM under different scenarios of background emissions (e.g., pre-industrial, modern) and forcings (e.g., historical, future).

## 5 Conclusions

We present the first application of a LIM to emulate the statistics and dynamics of a fully coupled CCM. The LIM is computationally lightweight (e.g., 100 years of simulation takes  $\sim$ 15 s) and can easily generate thousands of years of free-running simulations that reproduce the spectral and spatial statistics of the original simulation. As a linear system, the LIM can be decomposed into interpretable dynamical modes. These modes capture the expected chemistry-climate dynamics of ENSO, the dominant driver of variability in the pre-industrial chemical system, and they provide predictive skill when the LIM is used as a forecast model. When calibrated at seasonal resolution, the LIM produces skillful forecasts of chemical anomalies at lead times up to one year, independent of the atmospheric lifetime of the species.

The lightweight and modular nature of the LIM enables rapid hypothesis testing that would be expensive and cumbersome for a full-complexity CCM. Experiments that require CCM constraints on chemical variability can be tested with the LIM prior to or in lieu of the CCM. For example, we test the predictability of the chemistry-climate system in forecasts without the ENSO dynamical mode, highlighting ENSO's role as the dominant predictor of chemical variability: understanding how ENSO modulates chemical dynamics provides a useful constraint on predicting chemistry-climate interactions. These types of experiments would be difficult to implement in full CCMs, which require non-trivial modifications to climate dynamics in fully coupled simulations or to boundary conditions (e.g., SST, sea ice) in atmosphere-only simulations. Provided that the LIM (1) is fully damped, (2) reproduces the expected free-running statistics, and (3) contains dynamical modes are physically interpretable, it can be used to investigate chemistry-climate dynamics, variability, and predictability with initial value problems or free-running (equilibrium) simulations.

As a medium-complexity emulator of CCMs, the LIM also provides a baseline for evaluating higher-complexity data-driven emulators of CCMs or chemical transport models. These emulators should exceed the LIM's forecast skill and, if run dynamically, should capture or improve upon the coupled dynamics represented by the interpretable modes of the LIM.

A future application of the LIM enabled by its computational efficiency is coupled "online" data assimilation to reconstruct historical chemical variability. Previous work has used the LIM as a forecast model to reconstruct unobserved physical climate states over the past millennium (e.g., Perkins and Hakim, 2021). A LIM calibrated on coupled chemistry-climate dynamics can make use of both physical climate and chemical observations to jointly constrain past chemical states over unprecedented centennial timescales. Such reconstructions could correct chemical biases in CCM simulations that propagate from errors in the boundary conditions, parameterizations, or reaction schemes.

Code and data availability. The code and data for this study are available at https://doi.org/10.5281/zenodo.15829537. Code is also available at https://github.com/ericjmei/cclim-demo.

*Author contributions*. EJM, GJH, and AJT designed the research study; EJM, DS, and MT trained and tested empirical models and analyzed the results; EJM, GJH, and AJT wrote the manuscript.

Competing interests. No competing interests are present.

Acknowledgements. This work is part of the FETCH<sub>4</sub> project and is supported in part by Schmidt Sciences through the VESRI program. We gratefully acknowledge Paul Griffiths, who provided valuable feedback throughout this project that helped guide the direction of this work, and Vaishali Naik and Larry Horowitz, who provided the chemistry-climate model simulations used in this work.

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
