# Peer review of "Emulating chemistry-climate dynamics with a linear inverse model"

_EGUsphere, 2025_

## Author Comment (AC1)

We thank the reviewers for their useful questions and feedback. Reviewer comments have been re-printed in black, and our responses are in blue.

In addition to the changes to the Supplement noted below, the author list in the Supplement has been corrected to match the main text.

**Reviewer 1**

1. LIM has been widely and successfully used to predict internal climate variability, such as ENSO. However, its applicability becomes more limited when dealing with rapidly evolving external forcing. For atmospheric composition, external drivers such as anthropogenic and biomass burning emissions can undergo substantial short-term variations, which may challenge LIM's predictive skill. I suggest that the authors explicitly discuss LIM's strengths (e.g., robust performance for processes dominated by linear dynamics like ENSO) and limitations (e.g., reduced reliability under strongly time-varying external forcing) in the introduction and discussion sections. Such a discussion would provide a clearer perspective on the contexts in which LIM is most suitable for predicting atmospheric composition changes

Thank you for this insightful comment. We agree that external forcings on atmospheric chemistry could challenge the LIM's performance. The ability of the LIM to handle these forcings depends on 1) whether the dynamics (timescales and spatial patterns) of the forcings are well-separated from the resolved, linear dynamics of the chemistry-climate system during model calibration, and 2) whether the expected dynamical response to external forcings are well-represented in the resolved dynamics.

To investigate the above points, our planned future work involves model experiments to test the forecast and data assimilation reconstruction skill of the LIM under different scenarios of background emissions and forcings (e.g., pre-industrial vs. modern, historical transient simulations over industrialization, imperfect model experiments). These experiments will allow a better assessment of which forcings and scenarios a chemistry-climate LIM can accommodate.

We have added a few lines in the main text to incorporate these points:

Lines 48-50: The LIM has been used to understand the structure, evolution, and dynamical influence of climate modes such as ENSO (Penland and Sardeshmukh, 1995; Vimont et al., 2014, 2022) and the Pacific Decadal Oscillation (Alexander et al., 2008), where slow linear dynamics dominate.

Lines 323-331: Our predictability experiments assume a system with only slow-varying, natural processes (i.e., internal modes of variability), which we have shown can be captured by the LIM. For the chemical system, external forcings could be substantial (e.g., anthropogenic emissions during industrialization) and short-lived given the short atmospheric lifetime of many chemical species. A fundamental assumption of the LIM is a timescale separation between the linear, resolved dynamics and the fast-timescale, unresolved dynamics (Penland and Sardeshmukh, 1995). External forcings that have similar timescales or dynamics to the resolved dynamics could challenge LIM calibration, and external forcings with unexpected dynamics to the LIM's resolved dynamics could challenge its predictive skill. Future work should assess the performance of the chemistry-climate LIM under different scenarios of background emissions (e.g., pre-industrial, modern) and forcings (e.g., historical, future).

2. Figure 2a: It is not entirely clear whether the OH anomalies predicted by LIM and those simulated by GFDL-CM3 exhibit synchronous temporal variability or if they only match in terms of amplitude. Including a quantitative measure of agreement, such as the correlation coefficient between the two, would significantly enhance the evaluation of LIM's predictive capability.

The purpose of Figure 2a is to show that the timeseries simulated by the LIM and GFDL-CM3 are similar in magnitude and frequency statistics, which Figure 2b and Figure 2c show in more detail, respectively. Given the chaotic nature of the climate system and the different initial conditions used to perform the free-running simulations in the two models, we do not expect any synchronous temporal variability between the timeseries in Figure 2a. This point has been clarified in the main text:

Lines 173-174: Given the chaotic nature of the climate system, simulations from the LIM and GFDL-CM3 will have different trajectories but should be statistically indistinguishable.

3. Line 211-224: The authors show that LIM captures the variations of OH and ozone associated with the ENSO mode phase. It would strengthen this analysis to also present the corresponding ENSO-related OH and ozone responses simulated by GFDL-CM3. Such a comparison would allow a more direct assessment of LIM's ability to reproduce the underlying relationships captured by the full chemistry–climate model.

This is an excellent point. A common method to diagnose ENSO dynamics from an existing simulation is with a lead-lag analysis (e.g., Su et al. (2005)), in which the Niño 3.4 index is shifted in time and linearly regressed with variables in each grid cell of a simulation. We have performed this analysis with the withheld 1,000-year GFDL-CM3 test set, which is shown in Figure S25 in the revised Supplemental. Figure S25 can be directly compared with Figure 3, in which the patterns of growth (-8-month Niño 3.4 shift), peak (0-month Niño 3.4 shift), and decay (+5-month Niño 3.4 shift) of anomalies associated with El Niño match well with those in the ENSO mode. This is expected, since the linear dynamics of the LIM are calibrated with a linear regression of the lagged data.

A more robust method to ascertain ENSO dynamics in GFDL-CM3 would be to conduct new simulations. However, these experiments would be non-trivial and computationally expensive. A CCM simulation with no ENSO (or conversely, only ENSO) would require either: transient SSTs as boundary conditions in which dynamic components of ENSO have been entirely removed; or a method to remove ENSO while preserving other ocean dynamics in the ocean circulation model. Therefore, we feel that these simulations are beyond the scope of our work.

In our revised text, we add a comparison of the ENSO mode to the lead-lag analysis on the test set data, and we describe how future simulations could aid our analysis:

New Figure S25 in the supplemental:

[Figure]

Figure S25. Lead-lag analysis of Niño 3.4 index with SST (*top*), OH (*middle*), and ozone (*bottom*) in the withheld 1,000-year GFDL-CM3 simulation. Columns of the figure show the variables at each grid cell regressed against the Niño 3.4 index shifted 8 months earlier (*left*), with no time shifting (*middle*), and shifted 5 months later (*right*). Patterns are comparable to those in Figure 3 in the main text.

Lines 224-228: Coupled dynamics in the ENSO mode are supported by lead-lag analysis of the independent GFDL-CM3 test data (Figure S25), in which the Niño 3.4 index is shifted in time and linearly regressed with each variable. The patterns of El Niño growth, peak, and decay in the ENSO mode and the lead-lag analysis match for all variables. This is expected, since the deterministic dynamics of the LIM are calibrated with a linear regression of the time-lagged data.

Lines 231-232: Future CCM experiments could be performed to validate the ENSO chemistry-climate dynamics captured by the LIM.

4. Line 249: The manuscript uses the CE metric to evaluate LIM's forecast skill. Are there commonly accepted thresholds for interpreting CE values? For example, a 3-month OH forecast skill of 0.17 (figure 5), does this indicate a reasonably skillful forecast, or does it suggest limited predictive capability? Providing a brief explanation or reference regarding commonly accepted benchmarks for CE would help readers better assess the significance of the reported forecast skill values.

Thank you for this suggestion. The performance of the LIM is within expectations for forecasts of atmospheric variables, especially because skill has already been "lost" prior to the forecast due to EOF truncation. The 1-lag (seasonal) forecast skill of our LIM is comparable to the global average 2m air temperature reconstruction skill in Perkins and Hakim (2017) (CE≈0.2; Figure 8).

We have updated the text to include a point of comparison for the LIM performance in CE:

Lines 276-278: Forecast skill for chemical variables is lower than for SST but reasonable for prediction of atmospheric variables (e.g., comparable to 2m air temperature reconstruction skill in Perkins and Hakim, 2017): some chemical forecast skill remains for forecasts one year after initialization (OH CE decreases from 0.17 to 0.05 from 3 months to 1 year).

**Reviewer #2**

Line 180: Why does LIM performance fall off at the extreme ends of the frequency range? Is there a way to improve this? What are the implications of this for OH? Would this limit applicability of this methodology to trying to understand impacts of something like the Madden Julian Oscillation?

Thank you for these questions. We have made some improvements to Figure 2c to clarify the performance of the LIM at low and high frequencies:

1. Figure 2 compares simulations that are 1,000 years long. Therefore, we have truncated the lowest frequency (longest period) shown in Figure 2c to 1/500 cycles per year (500 years). The LIM's 95% confidence interval still fails to capture the lowest frequencies, but 1) these frequencies are poorly resolved given the length of the data, and 2) there are fewer data points on the low frequency end of the power spectrum compared to the high frequency.

2. We have added some transparency to the GFDL-CM3 power spectrum to allow better comparison with the 95% confidence interval from the LIM. With this change, it is clearer that the LIM's 95% confidence interval at high frequencies captures the broad features of the GFDL-CM3 power spectrum. In addition to this change, we also plot the power spectral density of individual ensemble members in S24 in the Supplemental Information, which show that the LIM produces the correct power at high frequencies.

Whether the LIM is well-suited to analyze chemical impacts of the Madden Julian Oscillation cannot be assessed here given the Nyquist frequency of the monthly-averaged data (6 cycles per year). Prior work using the LIM to analyze the MJO suggests that the LIM could be used in

this capacity (Cavanaugh et al., 2015), but higher temporal resolution would be required (e.g., daily resolution).

We have revised the text to reflect these changes:

Updated Figure 2c:

[Figure]

New Figure S24 in the Supplemental information:

[Figure]

Figure S24. Power spectral density of individual LIM ensemble members (orange) compared to that of the GFDL-CM3 simulation (blue, same as in Figure 2c in the main text).

Lines 180-185: The large 95% confidence interval of the power spectral density of the LIM simulations captures the broad features of the GFDL-CM3 spectrum. The GFDL-CM3 spectrum is noisy given its single realization, especially at high frequencies, but the power spectral density of individual LIM ensemble members show similar noisy characteristics (Figure S24). The LIM may have too little power at the lowest frequencies, but the low frequency portion of the GFDL-CM3 spectrum has few degrees of freedom and is poorly resolved given the length of the sample.

**References**

Cavanaugh, N. R., Allen, T., Subramanian, A., Mapes, B., Seo, H., & Miller, A. J. (2015). The skill of atmospheric linear inverse models in hindcasting the Madden–Julian Oscillation. *Climate Dynamics*, *44*(3), 897–906. https://doi.org/10.1007/s00382-014-2181-x

Perkins, W. A., & Hakim, G. J. (2017). Reconstructing paleoclimate fields using online data assimilation with a linear inverse model. *Climate of the Past*, *13*(5), 421–436. https://doi.org/10.5194/cp-13-421-2017

Su, H., Neelin, J. D., & Meyerson, J. E. (2005). Mechanisms for Lagged Atmospheric Response to ENSO SST Forcing. *Journal of Climate*, *18*(20), 4195–4215. https://doi.org/10.1175/JCLI3514.1